# Supramolecular copolymerization driven by integrative self-sorting of hydrogen-bonded rosettes

Keisuke Aratsu[1], Rika Takeya[1], Brian R. Pauw[2✉], Martin J. Hollamby[3✉], Yuichi Kitamoto[4], Nobutaka Shimizu[5], Hideaki Takagi[5], Rie Haruki[5], Shin-ichi Adachi[5] & Shiki Yagai[4,6✉]

Molecular recognition to preorganize noncovalently polymerizable supramolecular complexes is a characteristic process of natural supramolecular polymers, and such recognition processes allow for dynamic self-alteration, yielding complex polymer systems with extraordinarily high efficiency in their targeted function. We herein show an example of such molecular recognition-controlled kinetic assembly/disassembly processes within artificial supramolecular polymer systems using six-membered hydrogen-bonded supramolecular complexes (rosettes). Electron-rich and poor monomers are prepared that kinetically coassemble through a temperature-controlled protocol into amorphous coaggregates comprising a diverse mixture of rosettes. Over days, the electrostatic interaction between two monomers induces an integrative self-sorting of rosettes. While the electron-rich monomer inherently forms toroidal homopolymers, the additional electrostatic interaction that can also guide rosette association allows helicoidal growth of supramolecular copolymers that are comprised of an alternating array of two monomers. Upon heating, the helicoidal copolymers undergo a catastrophic transition into amorphous coaggregates via entropy-driven randomization of the monomers in the rosette.

[1] Division of Advanced Science and Engineering, Graduate School of Science and Engineering, Chiba University, 1-33 Yayoi-Cho, Inage-Ku, Chiba 263-8522, Japan. [2] BAM Federal Institute for Materials Research and Testing Unter den Eichen 87, 12205 Berlin, Germany. [3] School of Physical and Geographical Sciences, Keele University, Keele, Staffordshire ST55BG, UK. [4] Institute for Global Prominent Research (IGPR), Chiba University, 1-33 Yayoi-Cho, Inage-Ku, Chiba 263-8522, Japan. [5] Photon Factory, Institute of Materials Structure Science, High Energy Accelerator Research Organization, Tsukuba 305-0801, Japan. [6] Graduate School of Engineering, Chiba University, 1-33 Yayoi-Cho, Inage-Ku, Chiba 263-8522, Japan. ✉email: brian.pauw@bam.de; m.hollamby@keele.ac.uk; yagai@faculty.chiba-u.jp

The hierarchical self-assembly of macromolecular building blocks into hierarchical nanoarchitectures through the formation of oligomeric complexes is a strategy essential to living systems[1]. As small conformational changes in the complexes may severely affect the outcome of such hierarchical processes, biological molecular systems behave dynamically: they operate in conditions far from equilibrium and optimize component structures and functions according to environment[2]. This is true of not only the protein ensembles but also relatively simple one-dimensional assemblies of proteins, i.e., biological supramolecular polymers[3–7]. For example, two structurally related polypeptides of α- and β-tubulin monomer subunits complementarily form α–β heterodimer complexes through hydrophobic and electrostatic interactions[8–11]. These heterodimers then assemble one-dimensionally into microtubules. GTP hydrolysis within the tubulin heterodimer induces a small conformational change, triggering catastrophic microtubule disassembly[12]. We envisaged that the incorporation of a molecular recognition process to preorganize polymerizable complexes of an artificial supramolecular polymer[13–15] could generate smart macromolecular systems whose assembly and disassembly are controlled by the conformation of the complexes[16,17]. As synthetic polymers have evolved into functional materials with precise molecular sequences by controlling the kinetic reaction between monomers, similar efforts are essential in the development and practical application of supramolecular polymers.

We sought to create such an atypical artificial supramolecular polymer using simple synthetic molecules. In recent years, we have explored the self-assembly properties of a series of barbiturated π-conjugated molecules[18–22]. These molecules initially organize into six-membered complexes referred to as rosettes (Fig. 1a) through self-complementary hydrogen bonding. The rosettes subsequently stack through π–π interactions into a supramolecular polymer with an intrinsic radius of curvature of 7–15 nm. Extending this hierarchical process has also led to supramolecular polymers with helicoidal higher-order structures[21,22]. Here, we focus on molecule 1 (Fig. 1b)[23,24], which triggers the development of these unique supramolecular polymers. This molecule, whose π-conjugated core is naphthalene, mainly forms uniform toroidal (circular) fibers as closed nanostructures, rather than open-ended helicoidal structures (Fig. 1c). The efficient ring-closure during the continuous stacking of the rosette complex of 1 was attributed to its small enthalpic gain through π–π interactions, which prevents the supramolecular polymer chains from elongating into helicoidal structures. Previously, we have attempted to modulate the interaction between the rosettes by mixing a molecule with different substitution position or expanding the π-moiety[25–27]. However, mixtures of rosette-forming monomer subunits with others that contained a different π-conjugated core all exhibited narcissistic self-sorting[28–30].

Herein, we apply a much smaller mutation to 1, by introducing a carbonyl group in the linker moiety to form 2 (Fig. 1b). It is predicted that enhancing the stacking force between rosette complexes by admixing a monomer subunit with an electronically complementary π-conjugated core might realize alternating supramolecular copolymerization[31–34]: due to its modification, the π-conjugated core of 2 is electron-deficient, and can thus interact with 1 via electrostatic interactions. Mixtures of subunits 1 and 2 can randomly coassemble through hydrogen bonds at the rosette level, but at the same time, there may be an optimum unit arrangement that maximizes electrostatic interactions. We demonstrate that the competition of the random coaggregation of the two monomer subunits and the formation of an optimum rosette complex results in the autonomous progression of a supramolecular copolymerization process that selectively incorporates only one of the potential rosette complexes from the pool of various kinetically formed complexes over several days (Fig. 1e). The resulting thermodynamically stable helicoidal copolymers are thermally stable to a specific temperature, but above this transition catastrophically into a kinetic mixture of the rosette complexes due to the internal disorganization of the integrated rosette complex.

## Results

**Self-aggregation of 2.** The absorption maximum of monomeric 2 possessing the ester linkage in CHCl$_3$ is hypsochromically shifted by 23 nm compared to 1, reflecting the reduced electron density of its naphthalene core (Supplementary Fig. 12). Although a $^1$H NMR study of 2 in CDCl$_3$ indicated that its hydrogen-bonding capability was almost identical to 1, atomic force microscopy (AFM) and transmission electron microscopy (TEM) images of self-assembled 2 in methylcyclohexane (MCH) exhibited only indistinct linear fibrils (Fig. 1d and Supplementary Figs. 13–16). A temperature-dependent absorption spectral analysis of 2 in MCH showed a hypsochromic shift of the major absorption band upon cooling, while a bathochromic shift was observed for 1 (Supplementary Fig. 17). The temperature-dependence of the degree of aggregation could be fitted with a nucleation–elongation model[35,36]. A van't Hoff analysis of data collected at different concentrations provided an elongation enthalpy ($\Delta H° = -58$ kJ mol$^{-1}$) that is smaller than that of 1 ($\Delta H° = -72$ kJ mol$^{-1}$) as well as an elongation entropy ($\Delta S° = -92$ J mol$^{-1}$ K$^{-1}$ for 2, $\Delta S° = -139$ J mol$^{-1}$ K$^{-1}$ for 1) (Supplementary Fig. 18 and Supplementary Table 1). The very different spectroscopic and microscopic outcomes obtained upon introducing the ester group suggest that the lower electron density of the naphthalene core of 2 and/or dipole repulsion between the ester groups hamper the idiosyncratic stacking responsible for the formation of curved rosette aggregates.

**Kinetic coassembly.** Molecules of 1 and 2 kinetically coassembled into amorphous coaggregates when a 1:1 molar mixture of their monomers in MCH was cooled. Upon cooling a 1:1 mixture of 1 and 2 (hereafter denoted as 1/2) with a total concentration ($c_t$) of 100 μM from 80 to 20 °C at a rate of 1 °C min$^{-1}$, the absorption spectrum of the mixture broadened considerably (Fig. 2a and Supplementary Fig. 19). The increased absorption observed in the red-shifted region (450–500 nm) is characteristic of an electronic interaction between naphthalene chromophores. Due to the aggregation-induced suppression of bond rotation in the excited state, the mixture gradually became emissive in the yellow-green region upon cooling (Fig. 2b). The cooling curve obtained by plotting the absorption at 470 nm as a function of temperature confirmed the coassembly of 1 and 2: the onset of the spectral change of the mixture began at a higher temperature (~68 °C; blue dots in Fig. 2c) than that predicted by simulation that assumed narcissistic self-sorting (56 °C; black dots in Fig. 2c). The experimental curve was fitted using the nucleation–elongation model, and the values of $\Delta H°$ ($-55$ kJ mol$^{-1}$) and $\Delta S°$ ($-88$ J mol$^{-1}$ K$^{-1}$) were estimated by a van't Hoff analysis of the data collected at different concentrations (Supplementary Fig. 20). The $\Delta H°$ value is smaller than those of 1 ($-72$ kJ mol$^{-1}$) and 2 ($-58$ kJ mol$^{-1}$), suggesting that the coassembly did not exhibit any enthalpic advantage (Supplementary Table 1). Dynamic light scattering (DLS) measurements of the freshly cooled mixture showed an average hydrodynamic diameter ($D_H = 15$ nm) far smaller than observed for toroids of 1 and linear fibrils of 2 (Fig. 2d). In line with these observations, only amorphous coaggregates were observed in AFM and TEM images of the as-cooled fresh solution

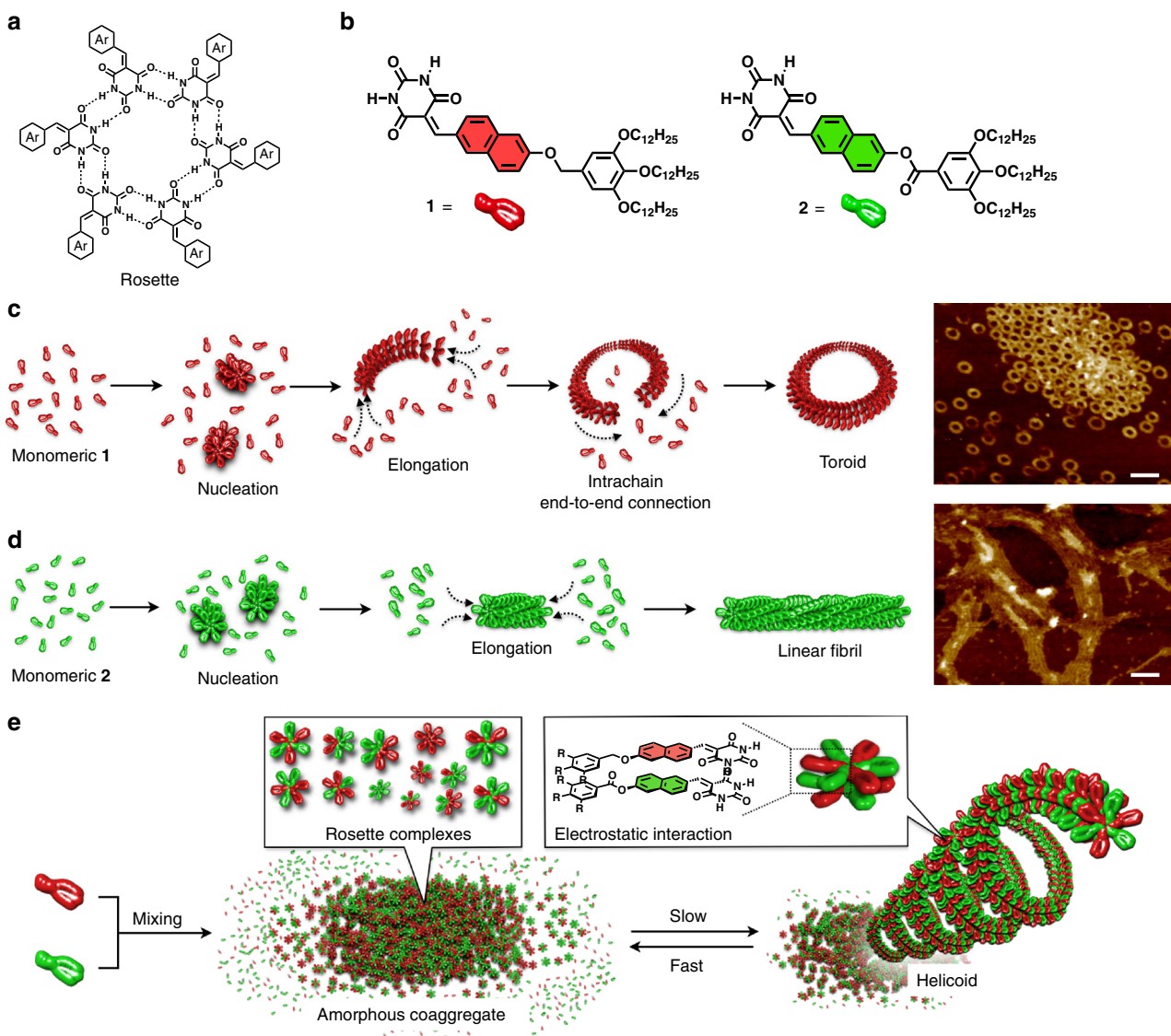

**Fig. 1 Supramolecular polymerization and copolymerization. a** Chemical structure of the barbituric acid rosette. **b** Chemical structure of toroid-forming molecule **1**, partially modified auxiliary molecule **2**. **c**, **d** Schematic representation of the supramolecular polymerization of **1** into a toroid (**c**) and **2** into a linear fibril (**d**) together with AFM images of the fibers ($c_t = 250\,\mu M$). Scale bars, 50 nm. **e** Schematic representation of the supramolecular copolymerization of **1** and **2** into a helicoid and the thermal transition from the helicoid to an amorphous coaggregate.

spin-coated onto appropriate substrates (Fig. 2f, h and Supplementary Fig. 21).

**Autonomous supramolecular copolymerization.** Isothermal supramolecular copolymerization of **1** and **2** occurs from the amorphous coaggregated state described above. The absorption spectra of the mixture gradually became vibronic over the course of several days at 20 °C, indicating a more uniform electronic interaction between the chromophores (Fig. 2a). A 50% attenuation of the emission was also observed (Fig. 2b) with a similar kinetic constant to that of the absorption spectral change ($k_{abs} = 0.095\,h^{-1}$; $k_{em} = 0.101\,h^{-1}$), suggesting a degree of charge transfer from electron-rich **1** to electron-poor **2** (Supplementary Fig. 22). These spectroscopic changes were accompanied by a gradual increase in aggregate size that reached an equilibrium value ($D_H = 190$ nm) after 60 h (Fig. 2e). When aliquots of the solution at different equilibration times were spin-coated onto highly oriented pyrolytic graphite (HOPG), the resulting AFM images demonstrated time-dependent growth of the helicoidal

structures (Fig. 2j–q and Supplementary Figs. 23 and 24). High-resolution imaging revealed an outer helicoidal diameter of approximately 23 nm and pitch of approximately 5.7 nm (Fig. 2g, i). The outer diameter is slightly larger than for toroids formed by **1** (20 nm), perhaps due to the flattening of the helicoids by edge-on adsorption onto the HOPG substrate. These results suggest that the topological extension from a toroidal to a helicoidal fiber was achieved[37], without synthetic extension of the aromatic core as we have done so previously[21,22].

Given that much slower supramolecular copolymerization kinetics were observed upon increasing $c_t$ (Supplementary Fig. 22), we inferred that the initially formed amorphous coaggregates are off-pathway products in the formation of the helicoids[38–40]. A time-dependent infrared (IR) analysis of the copolymerization revealed a remarkable shift for the two carbonyl vibrations of the hydrogen-bonded barbituric acids of **1** and **2** (Fig. 3a, b). Hence, it can be concluded that the two monomer subunits kinetically coassemble by hydrogen-bonding to form various rosette complexes, and continuously dissociate and reassemble with rearrangement of the hydrogen bonds until the enthalpy gain of

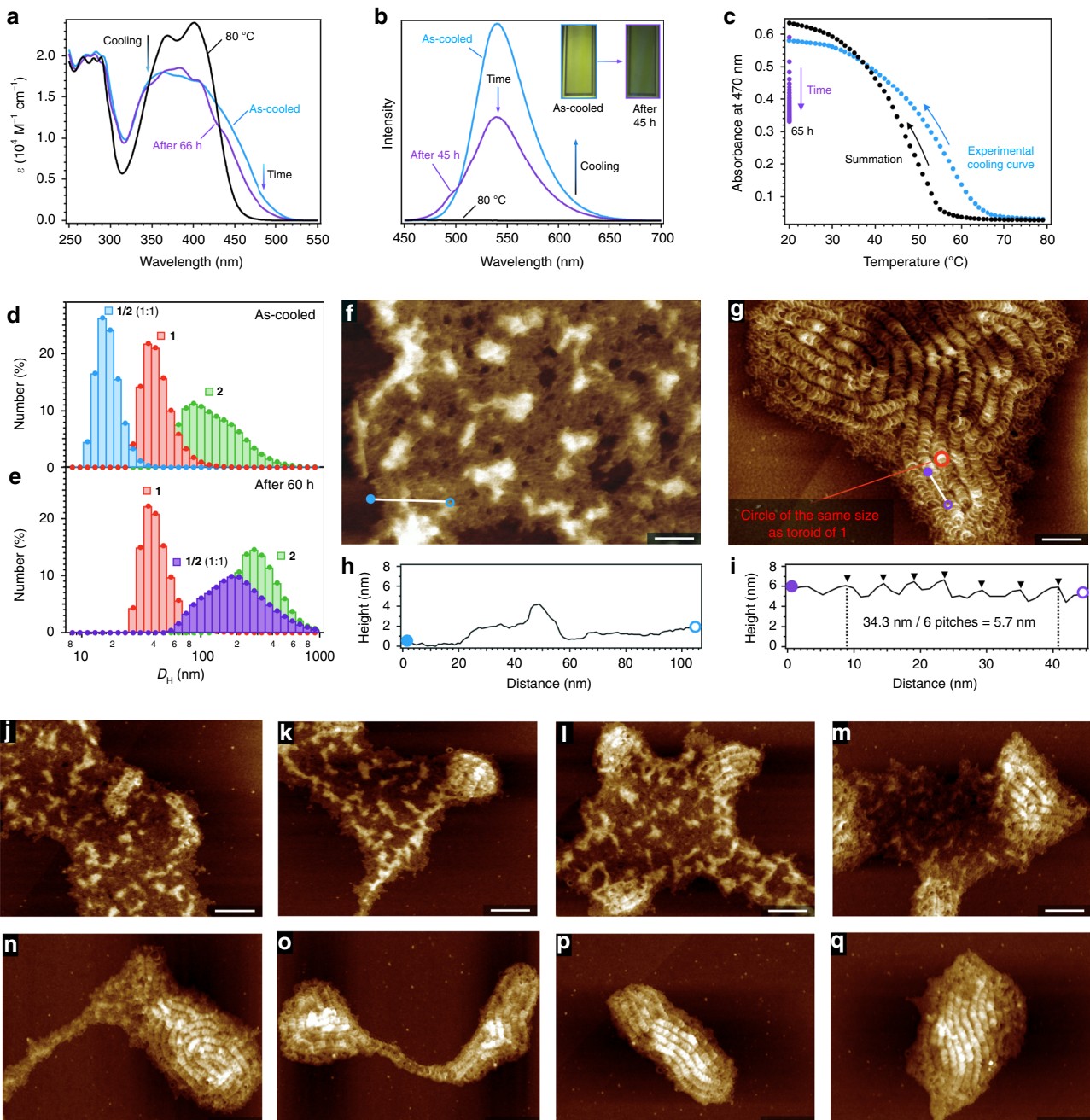

**Fig. 2 Supramolecular copolymerization. a, b** Absorption (**a**) and fluorescence emission (**b**) spectra of a 1:1 mixture of **1** and **2** ($c_t = 100\,\mu M$) at 80 °C (black lines) and 20 °C (blue lines) upon cooling, and the subsequent time-dependent change after several days at 20 °C (purple lines). Insets in **b** are photographs of the solution at 20 °C under illumination with UV light. **c** Plots of the change in the absorbance intensity at 470 nm as a function of temperature upon cooling (blue dots), followed by the time-dependent change (purple dots). The black dots were obtained by simple summation of the separately measured cooling plots of **1** ($c_t = 50\,\mu M$) and **2** ($c_t = 50\,\mu M$). **d, e** DLS profiles of **1** (red; $c_t = 100\,\mu M$), **2** (green; $c_t = 100\,\mu M$), and their 1:1 mixture (blue and purple; $c_t = 100\,\mu M$) in MCH measured immediately after cooling to 20 °C (**d**) and after aging for 60 h (**e**). **f, g** AFM images of the amorphous coaggregates (**f** aged for 0 h) and helicoids (**g** aged for 31 h) formed by a 1:1 mixture of **1** and **2** ($c_t = 100\,\mu M$) at 20 °C. **h, i** Cross-sectional analyses along the white lines in **f** and **g**, respectively. Scale bars, 50 nm. **j–q** AFM images showing the transition of the amorphous coaggregates into helicoids. The samples were spin-coated onto highly oriented pyrolytic graphite (HOPG) substrates and aged for 0.5 h (**j**), 1 h (**k**), 2 h (**l**), 4 h (**m**), 6 h (**n**), 20 h (**o**), 24 h (**p**), and 72 h (**q**). Scale bars, 100 nm.

the system is maximized through the electrostatic interaction between **1** and **2**. The ideal monomer arrangement to maximize the enthalpy gain is integrative self-sorting[41,42] into the alternating **1:2:1:2:1:2:** hydrogen-bonded rosette. These resulting complexes then stack with a rotational offset of ~60° to optimize the interaction between alternating **1** and **2** subunits within the complex (Fig. 1e).

Although the kinetic stability of the initially formed amorphous coaggregates is sufficiently low for their autonomous copolymerization to occur at 20 °C, this process could be further accelerated by the addition of a third component to the initial monomer mixture. Previously, we showed that **1** and its regioisomer **3** (Fig. 3c) narcissistically self-sort from kinetically formed amorphous coaggregates into toroidal and cylindrical

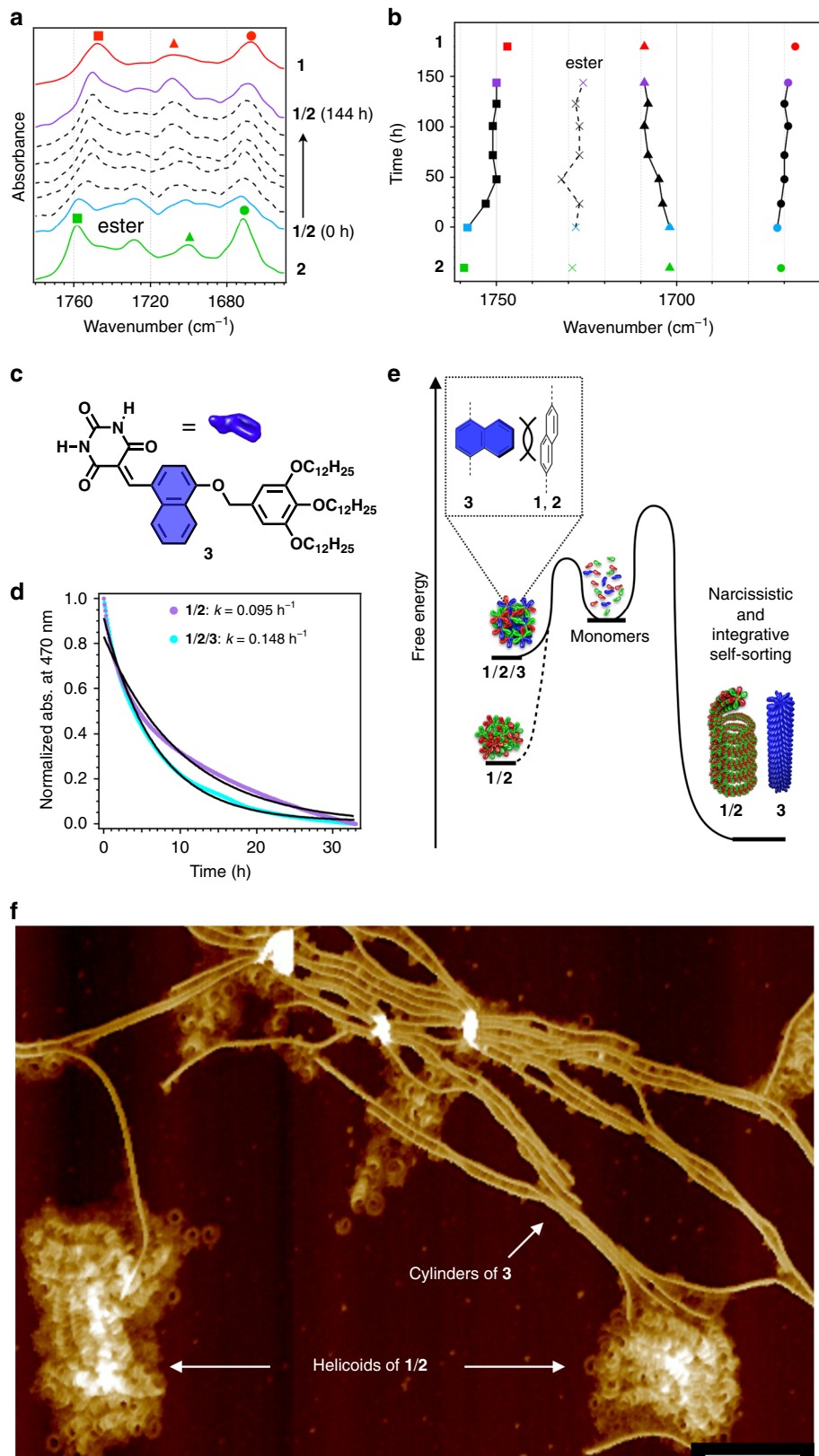

**Fig. 3 IR analysis and modulation of supramolecular copolymerization. a** IR spectra of toroids of **1** (red), linear fibrils of **2** (green), and a series of spectra showing the time-dependent spectral change of **1/2** ($c_t = 500\ \mu M$) from amorphous coaggregates (blue) to helicoids (purple). **b** Plots of the position of the maxima of the C=O stretching vibrations as a function of time (h). The colors and shapes of the symbols correspond to those in **a**. **c** Chemical structure of **3**. **d** Plots of the change in the normalized absorption at 470 nm for **1/2/3** (blue; $c_t = 150\ \mu M$) and **1/2** (purple; $c_t = 100\ \mu M$) at 20 °C after cooling. **e** Qualitative energy landscapes of the **1/2** and **1/2/3** systems. **f** AFM image of **1/2/3** after 24 h of aging. Scale bar, 100 nm.

supramolecular polymers, respectively[25]. This narcissistic self-sorting is due to the very different molecular conformations of these regioisomers; the naphthalene ring introduced on the short axis of **3** is sterically too demanding to allow the formation of intrinsically curved rosette stacks. We thus expected that the incorporation of this sterically demanding molecule could suppress the kinetic assembly of the various rosette complexes (Fig. 3e). Upon cooling a 1:1:1 ternary mixture of **1**–**3** (**1/2/3**; $c_t$ = 150 μM), only amorphous coaggregates were initially observed by AFM, with no trace of individual self-assemblies (Supplementary Fig. 25). An AFM image obtained after 1 day already showed not only the formation of helicoids of **1** and **2**, but also that of cylindrical fibers of **3**, suggesting the concerted progression of integrative and narcissistic self-sorting through dynamic molecular recognition (Fig. 3f). The kinetics of this ternary self-sorting process ($k_{abs}$ = 0.148 h⁻¹) were indeed much faster than that of the binary process ($k_{abs}$ = 0.095 h⁻¹; Fig. 3d). Similar control of kinetic effects through molecular recognition has been observed in other thermodynamic self-sorting systems[43].

**SAXS analysis of the supramolecular copolymerization**. The growth of the helicoidal supramolecular copolymers of **1** and **2** was studied by in situ small-angle X-ray scattering (SAXS) measurements. Figure 4a shows the time-dependent SAXS profiles of a concentrated 1:1 mixture ($c_t$ = 500 μM) of **1** and **2** measured after cooling. The scattered intensity, $I(Q)$ is plotted as a function of the scattering vector, $Q$. During the initial stages (0–2.5 h), the profiles correspond to small (<10 nm) size-disperse scatterers, with no visible oscillatory features. The observed increase in $I(Q)$ at low $Q$ (0.1–2 nm⁻¹), which is particularly apparent after 2.5 h, suggests the association of these early aggregates into larger objects and may indicate an onset of supramolecular copolymerization. From 5 h onward, a non-periodic oscillatory feature at $Q$ = 0.3–1 nm⁻¹ can be observed. Such features also appear in SAXS profiles arising from toroids of **1** and helicoids of previously reported **pnp** (Supplementary Figs. 26 and 27)[22,24]. As such, their presence here indicates the presence of supramolecular copolymers in solution. However, in comparison to that for **pnp**, here the SAXS from the helicoids of **1/2** exhibits an additional sharp scattering peak at $Q$ = 1.15 nm⁻¹ ($d$ = 5.5 nm), which notably increases in intensity over time (inset in Fig. 4a and Supplementary Fig. 28). This peak is likely to correspond to the helicoidal pitch ($p$) of the fibers as shown in Fig. 4g, which was estimated from AFM images to be ~5.7 nm (Fig. 2i).

To verify the above assignment of the SAXS features, the scattering of a model helicoid was simulated using the SPONGE, a materials science-oriented scattering pattern calculator (Supplementary Methods). SPONGE allows the simulation of a scattering pattern from any three-dimensional solid and permits the inclusion of realistic characteristics such as polydispersity and structural dynamics. By systematically varying individual structural parameters, including $p$ (Fig. 4b, g), the center-to-center distance ($D$; Fig. 4c, g), the persistence length ($n$) defined by the turn number (Fig. 4d), and the dynamical coil motion (spring motion; Fig. 4e), their effects on the scattering pattern have been identified. In Fig. 4a, we compare the experimental data with the closest simulation result obtained using $p$ = 5.5 nm, a number-weighted gaussian size dispersity of 5%, $D$ = 16 nm, a persistence length of 14 turns (160 nm) and, consistent with our previous work, an ellipsoidal wire cross-section of 6.0 × 3.6 nm² (flattened in the pitch direction). The overall shape of the simulation matches well with the experimental SAXS. Discrepancies in absolute intensity, particularly at mid-to-high $Q$, can be explained by (i) the unaccounted for presence of an incoherent background

in the experimental SAXS, and (ii) contrast effects (the simulations assume a two-phase sharp-contrast solvent-structure, whereas the helicoidal structures have a more complex internal electron-density structure).

The more apparent pitch peak in the experimental SAXS profile of the helicoids of **1/2** compared to that of the previously reported unimolecular helicoids of **pnp** is indicative of greater stiffness in the former. A reduction in stiffness would lead to increased swinging in the $x$ and $y$ directions as well as increased motion in the $z$-direction (spring motion; Fig. 4e). In SAXS analysis, such motion manifests as a reduced persistence length ($n$; Fig. 4d) of the fixed, uniform structure and will cause a sharp pitch peak to initially broaden and eventually become unobservable. Several reasons might account for a different stiffness: the elongation of the unimolecular helicoids of **pnp** was achieved by cooling, because the elongation process began at a high temperature in which the alkyl chains are in a molten state. This process is very different from that of the helicoids of **1/2**, for which the elongation proceeds at 20 °C. At this lower temperature, the long alkyl chains can assume more extended conformations, and may be interdigitated between loops (Fig. 4h). In fact, $p$ increased upon heating of the solution ($\Delta p$ = 0.5 nm, 20 → 50 °C) perhaps due to an enhanced molten state of the alkyl chains (Fig. 4f)[44].

**Distinct contribution of the monomer subunits**. Different supramolecular topologies of the homopolymers of **1** and **2** indicate that these monomer subunits may contribute differently to the formation of the helicoidal structure. To study this point, we examined the yield of helicoids at different mixing ratios of **1** and **2** while keeping $c_t$ constant at 100 μM. AFM analyses of the unequal mixtures after equilibration qualitatively demonstrated that helicoids are formed almost exclusively when an excess of **1** is used, whereas fewer were formed when **2** is present in excess (Fig. 5a, b and Supplementary Fig. 29). More quantitative data was obtained using SAXS. A plot of the relative intensity of the helicoidal pitch peak ($Q$ = 1.15 nm⁻¹) as a function of the **1**:**2** ratio showed that the intensity of this peak is considerably stronger for a 6:4 mixture (67% relative to the equimolar mixture) than for a 4:6 mixture (17% relative to the equimolar mixture; Fig. 5c, d). The same trend was observed for the relative absorption intensity at 470 nm, which is diagnostic of the electrostatic interaction between **1** and **2** (Supplementary Fig. 30). These results demonstrate that **1** is the structure-directing monomer subunit in the helicoidal supramolecular copolymerization, as it can form curved aggregates with itself, while **2** is an auxiliary monomer subunit that enhances the rosette–rosette interaction via its interaction with **1**. The decreased helicoid yield when **2** is in excess can be attributed to the increased probability of unfavorable self-stacking between **2** moieties.

The distinctly different contributions of **1** and **2** to the formation of helicoidal structures were further highlighted by experiments using chiral monomers. For that purpose, we synthesized the chiral subunits **1S** and **2S** (Fig. 5e and Supplementary Figs. 2 and 3), albeit that well-defined assemblies were not obtained from individual pure solutions of **1S** and **2S** (Supplementary Figs. 31 and 32). The sterically demanding chiral chains probably hamper the ordered stacking of their rosettes, as inferred from the observed quick precipitation upon cooling. We therefore investigated mixed systems of chiral and achiral monomers while keeping $c_t$ constant at 500 μM. As the curvature-forming monomer **1** is essential for the formation of helicoids, a 1:1 mixture of **1S/2** only afforded linear fibers (Supplementary Fig. 33). Even the combination of **1** and **1S** in a 1:1:2 mixture of **1S/1/2** did not produce helicoidal structures,

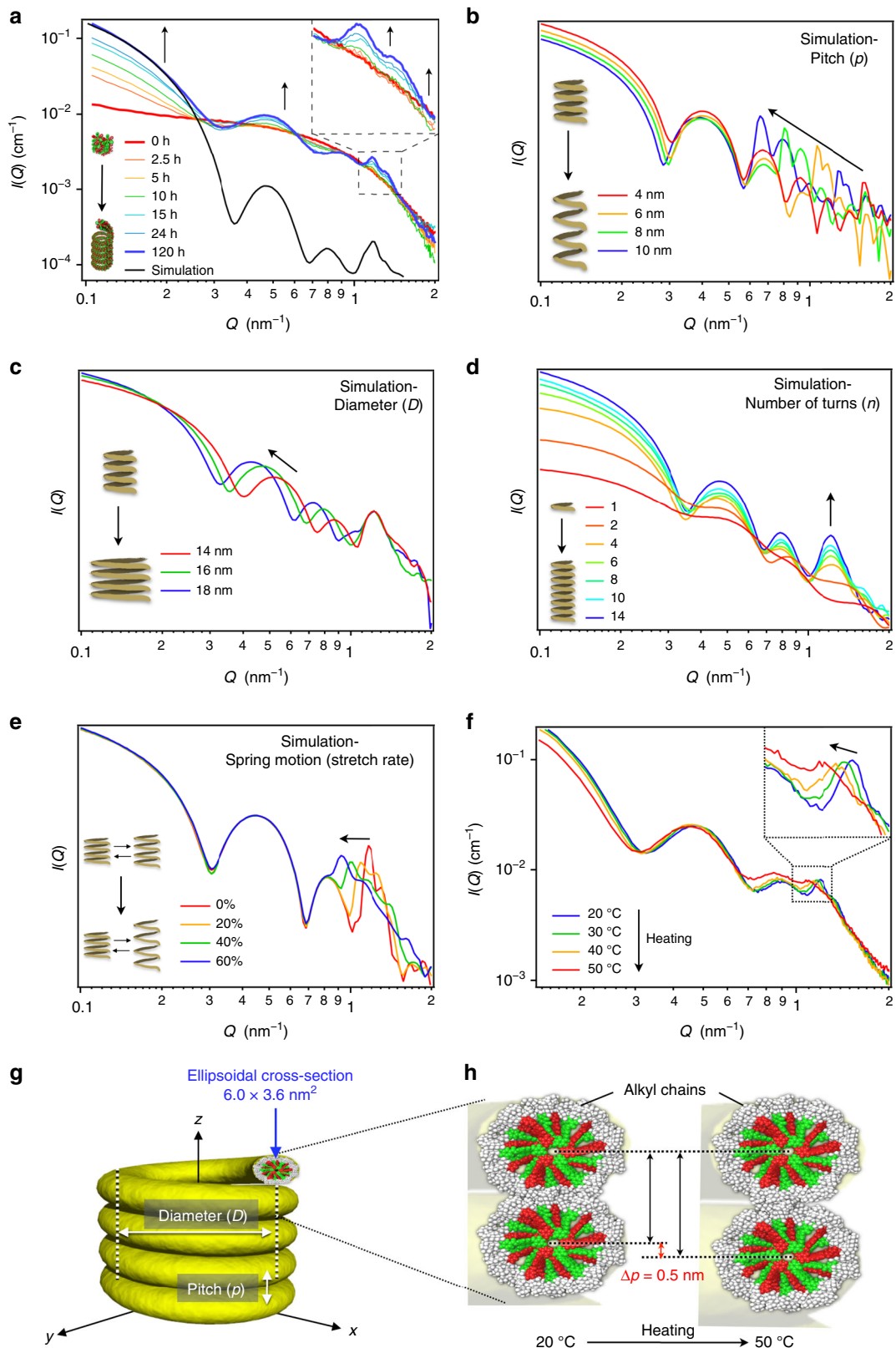

although amorphous coaggregates were afforded kinetically (Supplementary Fig. 34).

In contrast, the incorporation of chiral subunit **2S** into the system was found to promote the formation of chiral helicoids. Circular dichroism (CD) measurements demonstrated the time-dependent emergence of a Cotton effect for a 1:1 mixture of **1/2S**

after cooling; the mixture equilibrated after 2 h and afforded short coils (Supplementary Fig. 35). The Cotton effect observed for **1/2S** was entirely dissimilar to that of a 1:1 mixture of **2/2S**, suggesting that the chiral substituents of **2S** also control chiral packing of **1** through the alternative stacking of the monomers (Supplementary Fig. 36). Interestingly, when a fraction of **2S** was

**Fig. 4 Experimental and simulated SAXS profiles. a** Time-dependent SAXS profiles of **1/2** ($c_t = 500 \mu M$) during 120 h of aging. The arrows indicate changes with time. The black solid line represents the simulation result that was most similar to the experimental data, which is based on the following parameters: pitch distance ($p$) = 5.5 nm, coil center-to-center distance ($D$) = 16 nm, persistence length defined by the turn number ($n$) = 14, 5% dispersity, and dynamic motion ($p \pm 10\%$) in the $z$-direction. **b–e** Simulated SAXS profiles showing the effects of changing various parameters: **b** $p$ (for $D = 16$ nm, $n = 14$, and no dynamics); **c** $D$ (for $p = 5.5$ nm, $n = 14$, and no dynamics); **d** $n$ (for $p = 5.5$ nm, $D = 16$ nm, and no dynamics); **e** dynamics in the pitch direction (for $p = 5.5$, $D = 16$ nm, and $n = 14$). 5% dispersity in $D$ was included in all the simulations. **f** Temperature-dependent SAXS profiles of the helicoid of **1/2** at 20, 30, 40, and 50 °C. **g** Schematic representation of the helicoid showing the parameters $D$ and $p$ used for the model fitting of the SAXS data. **h** Schematic representation of the relaxing interdigitation of alkyl chains by the heating process.

replaced with achiral **2**, the Cotton effect of the resulting **1/2/2S** at equilibrium became more intense as the fraction of **2** was increased, and reached a maximum at a ratio of 2:1:1 (Fig. 5f, g). The intensity of the Cotton effect of this ternary mixture was twice that of the **1/2S** binary mixture, despite the reduced content of the chiral monomer **2S** (i.e., 125 μM in **1/2/2S** vs. 250 μM in **1/2S**)[45]. This optimum ternary mixture afforded elongated helicoidal fiber whose morphology was apparently the same as that of **1/2** (Fig. 5h). Although the helicity of the **1/2/2S** helicoids could not be identified due to their tightly folded structure, this negative Cotton effect strongly suggests the counter clockwise (CCW) rotation of rosettes that leads to $M$-type helicoids (Fig. 5i). The strong correlation between the Cotton effect and helicoidal structure corroborates our previous hypothesis that unidirectional rotation in the rosette stacking leads to the helicoidal growth of a series of supramolecular polymers[20].

**Catastrophic transition.** Although the reversibility of monomer binding is one of the major advantages of supramolecular polymers, the dissociation (depolymerization) process is not typically afforded much attention. In the solution state, most supramolecular polymers begin to depolymerize isodesmically upon heating, even if they are polymerized via a nucleation–elongation mechanism[46]. In this context, the thermal behavior of the present helicoids is unique as they abruptly transform into amorphous coaggregates upon heating due to the presence of kinetic coaggregates. Figure 6a shows the dissociation of the amorphous coaggregates and the helicoids of a 1:1 mixture of **1/2** ($c_t = 100$ μM) as monitored by UV–vis spectroscopy. The cooling and heating curves of the amorphous coaggregates (i and ii, respectively) exhibit a small thermal hysteresis, which originates from the progress of the competing supramolecular copolymerization process after cooling (iii). In sharp contrast, the heating curve of the helicoids exhibits a plateau up to 40 °C (iv), above which it gradually changes and completely overlaps with the cooling curve (i) without thermal hysteresis. When we switched the heating procedure to cooling at 55 °C, the subsequent second cooling curve was identical to the first (i), demonstrating the complete dissociation of the helicoids at ~50 °C (Supplementary Fig. 37). The aforementioned chiral 2:1:1 ternary mixture of **1/2/2S** also displayed a similar thermal response in the CD measurements: the Cotton effect was abruptly attenuated at ~40 °C and entirely eliminated at 55 °C (Fig. 6b).

In the plateau region (20–45 °C), DLS measurements showed only a moderate reduction in $D_H$ (200 nm → 157 nm; Fig. 6c). The $D_H$ abruptly decreases from 157 nm at 45 °C to 12 nm at 50 °C, and further to a reported 0.7 nm at 65 °C, indicating a complete dissociation to the monomeric state. The first abrupt decrease probably corresponds to the helicoids transitioning into amorphous coaggregates. This assertion was further supported by AFM and SAXS measurements (Fig. 6d, e and Supplementary Fig. 38).

The phase transition from helicoids to amorphous coaggregates suggests that the Gibbs free energy changes ($\Delta G°$) of the helicoids ($\Delta G°_{hel}$) and the amorphous coaggregates ($\Delta G°_{amo}$) should intersect

as the temperature is increased, due to the increasing contribution of entropy[47]. We therefore attempted to estimate the critical temperature ($T_c$) at which $\Delta G°_{amo} = \Delta G°_{hel}$ (Fig. 6f). The $\Delta G°$ values of the two species were estimated from temperature-dependent absorption measurements at different $c_t$. To observe the direct thermal dissociation of the helicoids into monomers, we diluted the solution to $c_t = 10$ μM, by which the onset of aggregation is expected to be sufficiently lower than the $T_c$. By this dilution, a sigmoidal dissociation of helicoids into monomers was observed with the onset of aggregation at 33 °C, suggesting dissociation via an isodesmic mechanism (Fig. 6g). A van't Hoff analysis of the isodesmic curve gave $\Delta H° = -96$ kJ mol$^{-1}$ and $\Delta S° = -218$ J mol$^{-1}$ K$^{-1}$ (Supplementary Fig. 39 and Supplementary Table 1). As described above, we obtained cooperative cooling curves for the amorphous coaggregates, from which we estimated values of $\Delta H° = -57$ kJ mol$^{-1}$ and $\Delta S° = -93$ J mol$^{-1}$ K$^{-1}$ using van't Hoff analysis. The larger negative entropy change of the helicoids clearly reflects the eventual conversion of the diverse rosette complexes into one alternating configuration (**1:2:1:2:1:2:**). This large entropic penalty could be compensated by the enthalpy change of $-96$ kJ mol$^{-1}$ in the helicoidal organization, i.e., the interaction between loops as well as the electrostatic interaction between **1** and **2**. From these $\Delta H°$ and $\Delta S°$ values, the $T_c$ at which $\Delta G°_{amo} = \Delta G°_{hel}$ was calculated to be 42 °C (Fig. 6f). This value can be rationalized by the observations that the helicoidal copolymerization proceeded when the temperature was kept at 35 °C during the cooling process, and the transition to the amorphous coaggregates proceeded when the temperature was kept at 45 °C during the heating process ($c_t = 100$ μM; Fig. 6h). Accordingly, the relative thermodynamic stability of the two species switches at ~40 °C, and the two processes are in equilibrium. Above 50 °C, however, the helicoid cannot form due to the entropic contribution, and therefore the heating curve coincided with the cooling curve.

## Discussion

Based on our hierarchical system for the construction of topological supramolecular polymers using hydrogen-bonded rosette complexes, we have presented a supramolecular copolymerization of two structurally near-identical monomer subunits. Due to the small structural difference between the two monomers, they kinetically coassemble to form various hydrogen-bonded complexes that associate in a disorderly manner into amorphous coaggregates. The weak electrostatic interactions between the two electronically complementary monomers drive the integration of the various complexes into one specific configuration through hydrogen bond rearrangement, which results in their supramolecular copolymerization into helicoidal superstructures. Notably, when homogeneous solutions of each of the monomers are prepared, one monomer predominantly forms thermodynamically stable toroidal assemblies, whereas the other can only afford ill-defined aggregates. In this context, we have successfully achieved the topological extension of toroidal to helicoidal architectures via supramolecular copolymerization. Although the helicoidal structures have a clear enthalpic advantage over amorphous coaggregates, the large entropic penalties associated with the

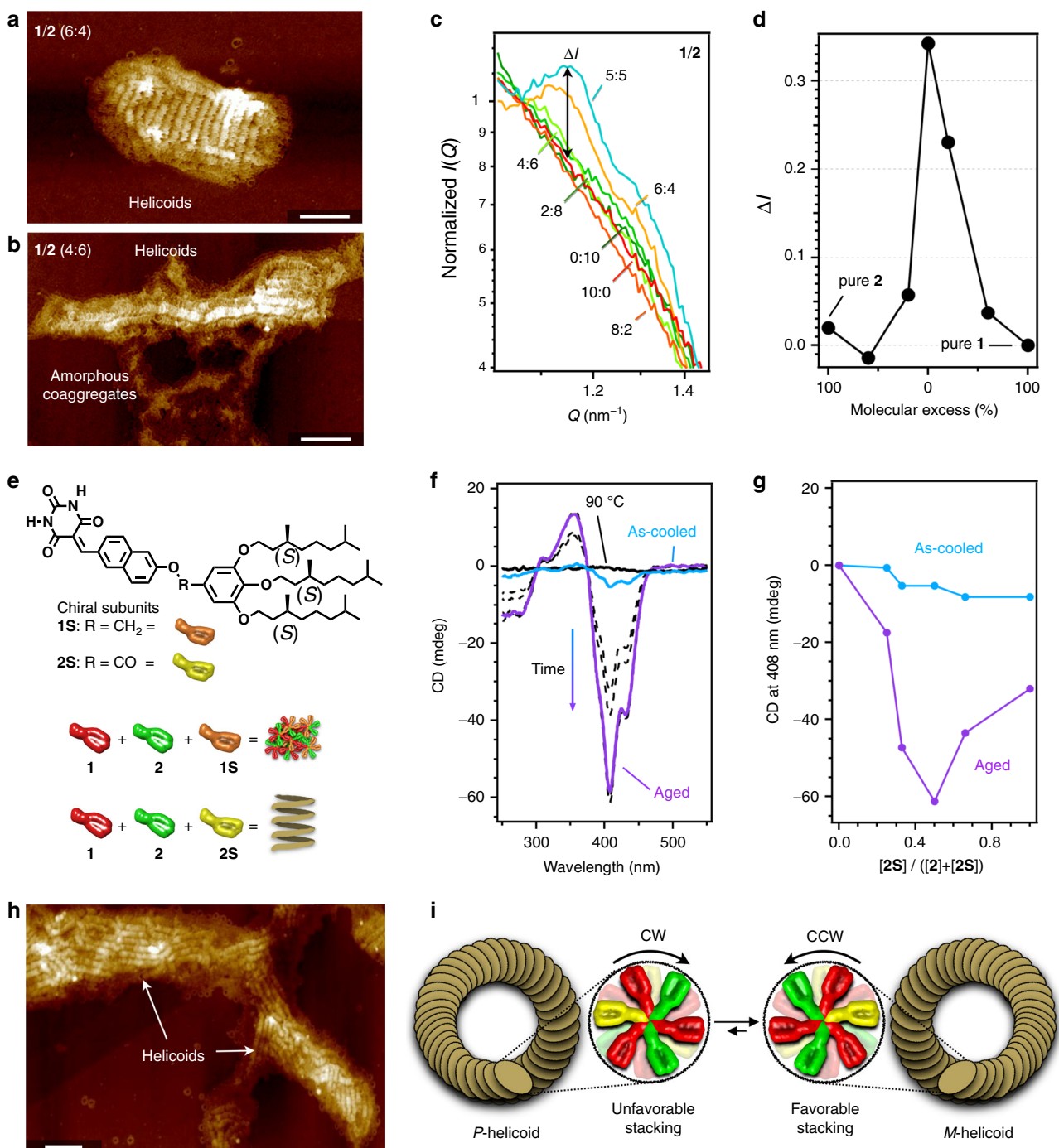

**Fig. 5 Distinct contributions of the different monomers. a, b** AFM images of **1/2** (6:4; **a**) and **1/2** (4:6; **b**) at $c_t = 100 \mu M$. The samples were prepared by spin-coating onto a HOPG substrate after 3 days of aging. Scale bars, 100 nm. **c** Normalized SAXS profiles at $Q = 1.06$ nm$^{-1}$ of mixtures of various ratios of **1** and **2**. Ratios: **1/2** (10:0; red), (8:2; dark orange), (6:4; orange), (5:5; blue), (4:6; light green), (2:8; green), and (0:10; dark green). **d** Plots of $\Delta I$ estimated from SAXS profiles as a function of the molecular excess. **e** Molecular structures of **1S** and **2S** as well as schematic representations of their tertiary mixtures. **f** CD spectra of **1/2/2S** (2:1:1; $c_t = 500 \mu M$) measured at 90 °C (black), immediately after cooling (blue), and after 9 days of aging (purple). **g** Plots of the intensity of the CD spectra at 408 nm immediately after cooling (blue) and after aging until reaching equilibrium (purple) as a function of the ratio of **2S** relative to the total amount of **2** derivatives; the concentration of **1** (250 μM) and the $c_t$ (500 μM) were kept constant. **h** AFM image of the helicoids of **1/2/2S** (2:1:1; $c_t = 500 \mu M$) after 7 days of aging. Scale bar, 100 nm. **i** Proposed mechanism for the elongation of the unidirectional helicoidal fiber.

alternating molecular arrays at both the hydrogen-bonded complex level and the hierarchical stacking level led to catastrophic transition behavior. All of these unique outcomes further highlight the advantages of designing supramolecular polymers in a hierarchical manner, which has rarely been achieved in artificial

supramolecular polymers, but is common in their natural counterparts. As demonstrated in this study, the hierarchical approach enables the integration of a molecular recognition process to preorganize the supramolecular building blocks, which is distinctly different from a supramolecular polymerization directly

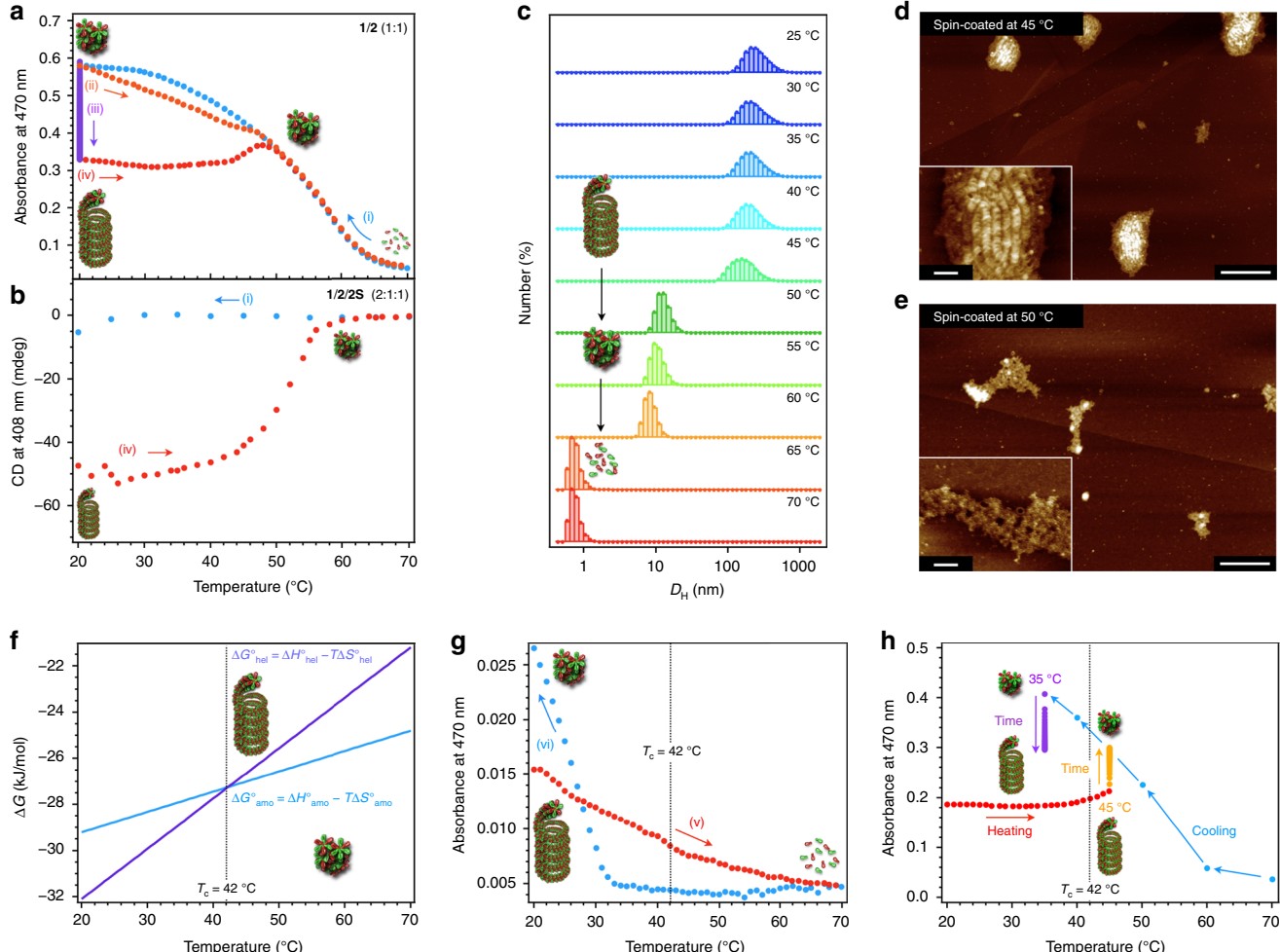

**Fig. 6 Catastrophic transition. a, b** Plots of absorbance at 470 nm (**a**) and CD intensity at 408 nm (**b**) as a function of temperature for **1/2** (1:1; $c_t = 100$ μM) and **1/2/2S** (2:1:1; $c_t = 500$ μM) in the first cooling process (i; blue), in the heating process immediately after cooling (ii; orange), in the aging process (iii; purple), and in the heating process after aging (iv; red). Heating and cooling rate is 1 °C min$^{-1}$. **c** Temperature-dependent DLS profiles of the helicoids of **1/2** (1:1; $c_t = 100$ μM) from 25 to 70 °C. **d, e** AFM images of **1/2** (1:1; $c_t = 100$ μM) samples prepared by spin-coating at 45 °C (**d**) and 50 °C (**e**). Scale bars, 300 nm and 50 nm (inset). **f** Phase diagram estimated from $\Delta G = \Delta H - T\Delta S$. **g** Plots of the spectral changes of the absorbance at 470 nm as a function of the temperature for **1/2** (1:1; $c_t = 10$ μM) during the heating (v; red) and cooling (vi; blue) processes. **h** Blue and purple dots: plots of absorbance at 470 nm as a function of temperature for **1/2** (1:1; $c_t = 100$ μM, amorphous coaggregate) in the first cooling process (blue) and subsequent aging process at 35 °C (10 h). Red and orange dots: plots of absorbance at 470 nm as a function of temperature for the equilibrated **1/2** (1:1; $c_t = 100$ μM, helicoid) in the heating process (red) and subsequent aging at 45 °C (10 h).

driven by host-guest systems. We are currently exploring further complex nanosystems featuring not only structural but also functional biomimicry, and the result of these studies will be reported in due course.

## Methods

**General**. Column chromatography was performed using 63–210 μm silica gel. All other commercially available reagents and solvents were of reagent grade and used without further purification. The solvents for the preparation of the assemblies were all spectral grade and used without further purification. $^1$H NMR spectra were recorded on Bruker DPS300 or JEOL JNM-ECA500 NMR spectrometers and chemical shifts are reported in parts per million (ppm) with the signal of TMS as internal standard. Electrospray ionization MS spectra were measured on an Exactive (Thermo Scientific). UV–vis and fluorescent spectra were recorded on a JASCO V660 spectrophotometer and a JASCO FP6600 spectropolarimeter, respectively. Both the spectrometers are equipped with Peltier device temperature-control unit. DLS measurements were performed on a Zetasizer Nano S (Malvern Instruments) device using non-invasive back-scatter technology (NIBS) under 4.0 mW He-Ne laser (633 nm). The scattering angle was set at 173°. Fourier transform IR (FT-IR) spectra were measured on JASCO FT/IR-4600 spectrometer.

**Materials**. Compound **1S**, **2**, and **2S** were synthesized according to the section of synthesis and characterization data in Supplementary materials. **1** and **3** were synthesized according to our previous report[23]. All starting materials and reagents were purchased from commercial suppliers and used without further purification. Air sensitive reactions were conducted under nitrogen atmosphere using dry solvents.

**Atomic force microscopy (AFM)**. AFM images were acquired under ambient conditions using Multimode 8 Nanoscope V microscope (Bruker Instruments) in peak force tapping (Scanasyst) mode. Silicon cantilevers (SCANASYST-AIR) with a spring constant of 0.4 N/m and frequency of 70 kHz (nominal value, Bruker, Japan) were used. Samples were prepared by spin-coating assembly solutions onto freshly cleaved HOPG.

**Small angle X-ray scattering (SAXS)**. SAXS experiments were performed at BL-10C of the Photon Factory of the High Energy Accelerator Research Organization (KEK) in Tsukuba, Japan[48]. Sample solutions were placed into 1.25-mm path length cells (20-μm thickness quartz glass windows surrounded by stainless steel), and the temperature was fixed at 293 K. X-ray with a wavelength of 1.5 Å and a sample-detector distance of 1029 mm (calibrated with silver behenate) resulted in a detectable $Q$ range in the order of 0.1–5.9 nm$^{-1}$. Sixty frames were collected with

each exposure time of 10 s. Because no radiation damage was observed, the collected data were averaged to give a total integration time of 600 s. The 2D scattering data (detector: DECTRIS PILATUS3 2M) were radially averaged to yield 1D scattering intensity data [$I(Q)$ vs. $Q$]. These data were then normalized using water as a reference, and the following subtraction of the background (quartz glass windows and solvent) gave absolute scattering intensity $I(Q)$ in cm$^{-1}$. All data reduction was performed using the software package SAngler[49]. In temperature-dependent SAXS experiments, the temperature of sample solution was controlled using a HCS302-LN190 (Instec Inc.). Sample solutions were placed in quartz capillaries (2 mm in diameter and 10 μm in wall thickness), and the top of the capillary was sealed with resin to avoid evaporation of the solvent at higher temperature. SAXS data were collected using X-ray with a wavelength of 1 Å.

## Data availability
All data needed to evaluate the conclusions in the paper are present in the paper and/or the Supplementary Materials. Additional data related to this paper may be requested from the corresponding authors.

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

## Acknowledgements
This work was supported by KAKENHI grant no. 26102010 and a Grant-in-Aid for Scientific Research on Innovative Areas "π-Figuration" (grant no. 26102001) from the Japanese Ministry of Education, Culture, Sports, Science, and Technology (MEXT). This work was performed with the approval of the Photon Factory Program Advisory Committee (proposal no. 2016G550). S.Y. acknowledges financial support from the Murata Science Foundation. K.A. thanks the JSPS for a research fellowship (17J02520). We thank Dr. Tomonori Ohba for assisting with TEM measurements.

## Author contributions

S.Y. and K.A. designed the project. K.A. synthesized **1**–**3**, and performed most of the experimental works except for **1S** and **2S**. R.T. synthesized **1S** and **2S**, and carried out the experimental works of them. N.S., H.T., R.H., and S.-i.A. collected the SAXS data. M.J.H. and B.R.P. simulated the SAXS data and wrote SAXS section of the paper. S.Y. and K.A. prepared the overall paper including figures. All authors including Y.K. have contributed by commenting on the paper. The overall project was directed by S.Y.

## Competing interests

The authors declare no competing interests.
