## [Peer Review File · Nature Communications]

Reviewers' comments:

Reviewer #1 (Remarks to the Author):

In this manuscript, Professor Yagai et al. report their research on a bioinspired supramolecular copolymerization system based on preorganized rosette complexes. The co-assembly of the electronically complementary monomers exhibits attractive autonomous copolymerization behavior controlled by the kinetic formation of various rosette complexes with different conformations. Both the structure of the copolymers and the mechanism of supramolecular copolymerization are supported substantially by experimental characterization as well as theoretical analysis. The copolymerization into hierarchical helicoids and the catastrophic transition into amorphous aggregates present a very interesting example for biomimic, molecular recognition-controlled kinetic self-assembly and disassembly processes.

Overall this excellent work is of high importance for advancement of supramolecular chemistry and the manuscript is written very well.

Minor revision is needed for publication:

(1) In page 5 and 6, the ΔH of 1, 2 and the coassembly was reported respectively as 63, 54, 57 KJmol⁻¹, and in page 13, ΔH value of 56.9 KJmol⁻¹ was reported for coassembly. Here, the number of significant digit for the ΔH should be unified. In addition, I suggest to report the ΔS values together with the ΔH values in the van't Hoff analysis results in page 5 and 6.

(2) In Page 13, 3rd paragraph, it is described that "we diluted the solution to $c_t = 10 \mu\text{M}$, at which the onset of aggregation (33°C) is below the T_c ". However, the value of T_c was obtained only after the measurement of ΔH and ΔS in page 14. The assertion of "below the T_c " should not appear too early.

(3) In Figure S28 in supporting information, for the inset, it is described "using the natural logarithm of the reciprocal c_t as a function of the reciprocal temperature" and in the figure the vertical coordinate is label as "ln K". This should be corrected.

Reviewer #2 (Remarks to the Author):

This manuscript by Yagai et al. describes the supramolecular copolymerization driven by integrative self-sorting of hydrogen-bonded rosettes. The method is very attractive and elegant since it uses a very simple approach to realize the molecular recognition in supramolecular polymers. I think the result is of sufficiently broad interest for Nature Communications. The manuscript was also a pleasure to read, with concepts and data explained clearly and in depth along with impressive figures. This paper has been among the best written ones during my recent reviews.

I have some comments which I believe are useful to improve the quality of the manuscript.

1. The area of supramolecular organic self-assembly has been developed for years. I acknowledge the contribution from this area to promote the understanding in molecular sciences. Of course, there are a great number of beautiful structures generated from supramolecular self-assembly. However, the eventual applications of this class of materials essentially rely on the bulk properties or the scalable fabrications. One may challenge the significance and the motivation of the current results by asking questions like: Why do we need this kind of research except that we can get pretty microscopy images? I suggest that the authors improve the introduction to justify themselves better.

2. If the authors acknowledge that their materials are termed as "polymers", they should realize that how polymers can perform superior over other materials. As the monomers are polymerized, the collective behavior shown in the polymers render many useful properties.

3. I would ask the authors whether they can control the assembly in a scalable way, since they got the materials in a little amount only for characterizations. And if they can prepare the bulk materials of, for instance, toroids or fibrils, can they align the assemblies into macroscopic ordered structures. If you just get some nice morphologies from AFM or TEM, the aspect of applications will be weakened. Or can you really incorporate the toroids or fibrils into some other mediums to build composites? If you said your systems have mimicked the biological systems, you should not forget that in reality they are related to very complicated environments.

4. Can the authors discuss what factors determine the diameter of the helicoid? Are there any elastic energy relationships responsible for the structure?

5. The plot of Figure 4a is poorly presented. Please change the colors and the style of the lines to make sure that the data are clear to understand. –Also Figure 5c and d are too small to see.

Reviewer #3 (Remarks to the Author):

The manuscript by Aratsu et al. reports on the exciting and really interesting co-assembly features of two complementary naphthalene-based barbiturates endowed with electron-donating (ether) and electron-withdrawing (ester) functional groups. The manuscript collects a very complete set of experiments, microscopy images and spectroscopic measurements, to justify the experimental evidences found out in the course of such investigations. As in many other publications, as a typical feature of the research group, it is worthy to mention the quality of the AFM images. At the same time, I have to remark the amount of experimental evidences provided to demonstrate the hypothesis of the manuscript.

I consider that the manuscript meets the quality criteria to be published in a reputed journal like Nature Comm. However, there are some points that should be addressed prior to its publication.

1) As stated before, the authors naphthalene-based barbiturates endowed with electron-donating (ether) and electron-withdrawing (ester) functional groups. It is obvious that the electronic nature of the ether or ester functional groups is opposite. However, I am very curious about the electronic complementarity of these two functional groups. It would be very useful if the authors could provide any experimental or bibliographic reference about the redox potentials of such compounds that could demonstrate the potential charge transfer effect between these moieties, as it is mentioned in page 6 of the manuscript.

2) In a manuscript dealing with co-assembly, it is very important to derive the thermodynamic parameters associated to the supramolecular polymerization of both the pristine components and also to the investigated mixture of components. The authors have made such calculations and provide the corresponding parameters. However, there are two circumstances that make these values not very accurate. The first one is the fact the authors have utilized the model described by Meijer et al. in 2006 (reference S9 in the Supporting Information). At the same time, and more importantly, the authors have utilized only three curves to derive these parameters. Of course, if you have three point the R² value for the linear fitting is 1. The authors should: a) make the calculations with, at least, four cooling curves or b) utilize the most recent model published by ten Eikelder and coworkers (J. Phys. Chem. B, 2012, 116, 5291) that allows a global fitting of the data and, consequently, a more accurate determination of the parameters.

3) The authors report on a very complex system composed by three components, in certain cases one of them is chiral. It is absolutely necessary to perform the corresponding studies with binary systems. For instance, how are the binary 1/2 or 2/3 systems behaving? More interesting is the

ternary systems incorporating the chiral congener. In this case, the mixtures 1+1S or 2+2S would be examples of sergeant-and-soldiers experiments that provide very useful information about the required ratio of the chiral sergeant, and also the energetics derived of this chiral amplification phenomenon, to further optimize the investigation of the ternary systems.

Overall, I recommend the publication of this timely work in Nature Comm. after major revision.

Answer to the comments raised by Reviewer 1 **(Additions to the main text are highlighted yellow)**

General Comments:

In this manuscript, Professor Yagai et al. report their research on a bioinspired supramolecular copolymerization system based on preorganized rosette complexes. The co-assembly of the electronically complementary monomers exhibits attractive autonomous copolymerization behavior controlled by the kinetic formation of various rosette complexes with different conformations. Both the structure of the copolymers and the mechanism of supramolecular copolymerization are supported substantially by experimental characterization as well as theoretical analysis. The copolymerization into hierarchical helicoids and the catastrophic transition into amorphous aggregates present a very interesting example for biomimic, molecular recognition-controlled kinetic self-assembly and disassembly processes.

Overall this excellent work is of high importance for advancement of supramolecular chemistry and the manuscript is written very well. Minor revision is needed for publication:

Author's Response: Thank you very much for your in-depth understanding of our study and the insightful comments.

Specific Comments (1):

In page 5 and 6, the ΔH of 1, 2 and the coassembly was reported respectively as 63, 54, 57 KJmol⁻¹, and in page 13, ΔH value of 56.9 KJmol⁻¹ was reported for coassembly. Here, the number of significant digit for the ΔH should be unified. In addition, I suggest to report the ΔS values together with the ΔH values in the van't Hoff analysis results in page 5 and 6.

Author's Response to the Specific Comments (1): We thank the reviewer for indicating discrepancies in significant digit. We fixed those discrepancies as follows. Please note that the new H° values are different from those given in the original manuscript, because we also responded to the comment of the reviewer 3. For the same reason, we added the elongation entropy values.

Page 5 and 6

Original:

A van't Hoff analysis of data collected at different concentrations provided an elongation

enthalpy ($H^\circ = -54 \text{ kJ mol}^{-1}$) that is smaller than that of **1** ($H^\circ = -63 \text{ kJ mol}^{-1}$) (Supplementary Fig. 7; Supplementary Table 1).

Revised:

A van't Hoff analysis of data collected at different concentrations provided an elongation enthalpy ($H^\circ = -58 \text{ kJ mol}^{-1}$) that is smaller than that of **1** ($H^\circ = -72 \text{ kJ mol}^{-1}$) as well as an elongation entropy ($S^\circ = -92 \text{ J mol}^{-1} \text{ K}^{-1}$ for **2**, $S^\circ = -139 \text{ J mol}^{-1} \text{ K}^{-1}$ for **1**) (Supplementary Fig. 7 and Supplementary Table 1).

Original:

The experimental curve was fitted using the nucleation-elongation model, and the value of H° (-57 kJ mol^{-1}) was estimated by a van't Hoff analysis of the data collected at different concentrations (Supplementary Fig. 9). This value lies between the H° values of **1** (-63 kJ mol^{-1}) and **2** (-54 kJ mol^{-1}), suggesting that the coassembly did not exhibit any enthalpic advantage (Supplementary Table 1).

Revised:

The experimental curve was fitted using the nucleation-elongation model, and the values of H° (-55 kJ mol^{-1}) and S° ($-88 \text{ J mol}^{-1} \text{ K}^{-1}$) were estimated by a van't Hoff analysis of the data collected at different concentrations (Supplementary Fig. 9). The H° value is smaller than those of **1** (-72 kJ mol^{-1}) and **2** (-58 kJ mol^{-1}), suggesting that the coassembly did not exhibit any enthalpic advantage (Supplementary Table 1).

Page 12

Original:

A van't Hoff analysis of the isodesmic curve gave $H^\circ = -95.6 \text{ kJ mol}^{-1}$ and $S^\circ = -217.8 \text{ J mol}^{-1} \text{ K}^{-1}$ (Supplementary Fig. 28; Supplementary Table 1). As described above, we obtained cooperative cooling curves for the amorphous coaggregates, from which we estimated values of $H^\circ = -56.9 \text{ kJ mol}^{-1}$ and $S^\circ = -93.0 \text{ J mol}^{-1} \text{ K}^{-1}$ using van't Hoff analysis.

Revised:

A van't Hoff analysis of the isodesmic curve gave $H^\circ = -96 \text{ kJ mol}^{-1}$ and $S^\circ = -218 \text{ J mol}^{-1} \text{ K}^{-1}$ (Supplementary Fig. 28 and Supplementary Table 1). As described above, we obtained

cooperative cooling curves for the amorphous coaggregates, from which we estimated values of $\Delta H^\circ = -57 \text{ kJ mol}^{-1}$ and $\Delta S^\circ = -93 \text{ J mol}^{-1} \text{ K}^{-1}$ using van't Hoff analysis.

Page 13

Original:

This large entropic penalty could be compensated by the enthalpy change of $-95.6 \text{ kJ mol}^{-1}$ in the helicoidal organization, i.e., the interaction between loops as well as the electrostatic interaction between **1** and **2**.

Revised:

This large entropic penalty could be compensated by the enthalpy change of -96 kJ mol^{-1} in the helicoidal organization, i.e., the interaction between loops as well as the electrostatic interaction between **1** and **2**.

Specific Comments (2):

In Page 13, 3rd paragraph, it is described that "we diluted the solution to $c_t = 10 \mu\text{M}$, at which the onset of aggregation (33°C) is below the T_c ". However, the value of T_c was obtained only after the measurement of ΔH and ΔS in page 14. The assertion of "below the T_c " should not appear too early.

Author's Response to the Specific Comments (2): Thank you very much for this important suggestion. According to the suggestion, we have deleted the corresponding part.

Page 12

Original:

To observe the direct thermal dissociation of the helicoids into monomers, we diluted the solution to $c_t = 10 \mu\text{M}$, at which the onset of aggregation (33°C) is below the T_c . A sigmoidal dissociation of helicoids into monomers was observed, suggesting dissociation via an isodesmic mechanism (Fig. 6g).

Revised:

To observe the direct thermal dissociation of the helicoids into monomers, we diluted the solution to $c_t = 10 \mu\text{M}$, by which the onset of aggregation is expected to be sufficiently lower than the T_c . By this dilution, a sigmoidal dissociation of helicoids into monomers was observed with the onset

of aggregation at 33 °C, suggesting dissociation via an isodesmic mechanism (Fig. 6g).

Specific Comments (3):

In Figure S28 in supporting information, for the inset, it is described "using the natural logarithm of the reciprocal c_t as a function of the reciprocal temperature" and in the figure the vertical coordinate is label as "ln K". This should be corrected.

Author's Response to the Specific Comments (3): We apologize for our mistake. We fixed the errors as follows.

Supplementary Figure 28 and its caption

Original:

Inset is van't Hoff plots using the natural logarithm of the reciprocal c_t as a function of the reciprocal temperature (T). Purple line shows a linear fitting.

Revised:

Inset shows the corresponding van't Hoff plot obtained by plotting natural logarithm of equilibrium constant (K_{iso}) as a function of T^{-1} . Purple line shows a linear fitting.

Answer to the comments raised by Reviewer 2 **(Additions to the main text are highlighted green)**

General Comments:

This manuscript by Yagai et al. describes the supramolecular copolymerization driven by integrative self-sorting of hydrogen-bonded rosettes. The method is very attractive and elegant since it uses a very simple approach to realize the molecular recognition in supramolecular polymers. I think the result is of sufficiently broad interest for Nature Communications. The manuscript was also a pleasure to read, with concepts and data explained clearly and in depth along with impressive figures. This paper has been among the best written ones during my recent reviews. I have some comments which I believe are useful to improve the quality of the manuscript.

Author's Response: We are pleased with the reviewer's great comment, and we are especially impressed by the comment "This paper has been among the best written ones during my recent reviews."

Specific Comments (1):

The area of supramolecular organic self-assembly has been developed for years. I acknowledge the contribution from this area to promote the understanding in molecular sciences. Of course, there are a great number of beautiful structures generated from supramolecular self-assembly. However, the eventual applications of this class of materials essentially rely on the bulk properties or the scalable fabrications. One may challenge the significance and the motivation of the current results by asking questions like: Why do we need this kind of research except that we can get pretty microscopy images? I suggest that the authors improve the introduction to justify themselves better.

Author's Response to the Specific Comments (1): We believe that all researchers engaging in this research field should consider and possibly address similar questions raised by the reviewer because there is a significant gap between practical applications and beautiful structures assembling in micromolar concentration solutions. Researches that merely find new phenomena and beautiful structures seem to no longer contribute significantly to the progress of this research field except for educational purposes. So, in this paper, we have focused on the principles behind such a beautiful structure. As the reviewer has pointed out, the development

of soft materials for practical use is intended to ensure that they can work in the bulk state. However, these designed main chains are realized only by controlling the dynamic reactions between monomers in solution. For example, polymerization of styrene and maleic anhydride results in an alternating copolymer due to the large difference of e values (which is an indication of monomer's polarity). Without an understanding of the underlying mechanism, such a functional copolymer would not have been realized. The same is true of the evolution of supramolecular polymers. As such, in response to the reviewers' excellent suggestion, we added the following sentence in order to justify why this kind of study is important.

Page 3

Added sentence:

As synthetic polymers have evolved into functional materials with precise molecular sequences by controlling the kinetic reaction between monomers, similar efforts are essential in the development and practical application of supramolecular polymers.

Specific Comments (2):

If the authors acknowledge that their materials are termed as “polymers”, they should realize that how polymers can perform superior over other materials. As the monomers are polymerized, the collective behavior shown in the polymers render many useful properties.

Author's Response to the Specific Comments (2): In our recent series of works, we have not yet find the superior properties except for uniqueness in the polymerization process and the nanostructure. However, we have already started applied researches, and we believe that the emergence of unprecedented soft materials featuring unique nanoscale topologies and dynamic monomer-aggregate equilibrium will be realized in the future.

Specific Comments (3):

(i) I would ask the authors whether they can control the assembly in a scalable way, since they got the materials in a little amount only for characterizations. And if they can prepare the bulk materials of, for instance, toroids or fibrils, can they align the assemblies into macroscopic ordered structures. If you just get some nice morphologies from AFM or TEM, the aspect of applications will be weakened. (ii) Or can you really incorporate the toroids or fibrils into some other mediums to build composites? (iii) If you said your systems have mimicked the biological

systems, you should not forget that in reality they are related to very complicated environments.

Author's Response to the Specific Comments (3): Thank you for your important comments.

(i) Simplicity of our molecules (barbiturated π -conjugated compounds) allow us to prepare bulk amount (in a gram scale). Although not in a gram scale, we have observed the same phenomenon at $c_t = 0.02$ M (Please see the AFM images shown below). Furthermore, we have observed our topological supramolecular polymers under such a condensed condition tend to assemble orderly to show some mesomorphic (e.g., lyotropic) behavior. Because these topics is beyond the scope of the present work, we will report them in our future publications.

(ii) This is exactly one of the important thematic issues in our group. Now we are exploring a possibility to make composites a series of our well-defined supramolecular polymers with MOF and other polymeric materials to shift our researches to more non-solvent or confined solution phases.

(iii) Biomimicry is a big motivation in supramolecular chemistry, but at the same time is very challenging to design pragmatic systems. We believe that extracting essential principles of interacting molecules from biological systems pave the way to elevate the standard of artificial supramolecular systems that show life-like behavior in simple material systems such as gels and liquid crystals, both of which lack time-evolving dynamic behaviors. Systems chemistry, chemical fuel-driven non-equilibrium dissipative systems, precise supramolecular polymerization, and related multicomponent self-assembly systems are accordingly emerging new topics in this research field, and we hope the present work with a special emphasis on the marriage of two different supramolecular events, i.e., self-sorting and supramolecular polymerization, will motivate researchers to explore more complex multicomponent supramolecular materials.

Specific Comments (4):

Can the authors discuss what factors determine the diameter of the helicoid? Are there any elastic energy relationships responsible for the structure?

Author's Response to the Specific Comments (4): Given the various monomers we have synthesized and the supramolecular polymer topologies they form, the stacking arrangement of rosettes is more responsible to determine the diameter of intrinsic curvature. This notion is clearly supported from the observation that the degree of curvature in toroid and helicoid of one monomer is almost identical. However, there should be an interesting mechanical property of the helicoid along to its long axis, namely spring motion. We are planning to study such a property by using probe microscopy, and hopefully able to update our progress in the near future.

Specific Comments (5):

The plot of Figure 4a is poorly presented. Please change the colors and the style of the lines to make sure that the data are clear to understand. –Also Figure 5c and d are too small to see.

Author's Response to the Specific Comments (5): According to the reviewer's opinion, we improved the appearance of Figure 4a as follows.

Original Figure 4a:

Revised Figure 4a:

Original Figure 5:

Revised Figure 5:

Answer to the comments raised by Reviewer 3 (Additions to the main text are highlighted blue)

General Comments:

The manuscript by Aratsu et al. reports on the exciting and really interesting co-assembly features of two complementary naphthalene-based barbiturates endowed with electron-donating (ether) and electron-withdrawing (ester) functional groups. The manuscript collects a very complete set of experiments, microscopy images and spectroscopic measurements, to justify the experimental evidences found out in the course of such investigations. As in many other publications, as a typical feature of the research group, it is worthy to mention the quality of the AFM images. At the same time, I have to remark the amount of experimental evidences provided to demonstrate the hypothesis of the manuscript.

I consider that the manuscript meets the quality criteria to be published in a reputed journal like Nature Comm. However, there are some points that should be addressed prior to its publication.

Author's Response: Thank you very much for the kind comments and remarks and the in-depth understanding of our manuscript. All the comments raised by this reviewer were addressed in the revised manuscript.

Specific Comments (1):

As stated before, the authors naphthalene-based barbiturates endowed with electron-donating (ether) and electron-withdrawing (ester) functional groups. It is obvious that the electronic nature of the ether or ester functional groups is opposite. However, I am very curious about the electronic complementarity of these two functional groups. It would be very useful if the authors could provide any experimental or bibliographic reference about the redox potentials of such compounds that could demonstrate the potential charge transfer effect between these moieties, as it is mentioned in page 6 of the manuscript.

Author's Response to the Specific Comments (1): Because we cannot measure the redox potential of these molecules in our group due to lack of suitable facilities, we do not have any information about it at present. However, we believe that the difference of redox potential in the present system is insufficient to form a charge-transfer complex in this system due to no CT transition with absorption spectra. Instead of the measurement for redox potential, we performed DFT calculations of **1'** and **2'** removed the long alkyl chains. The DFT calculation showed slightly different HOMO–LUMO gaps between the two compounds, suggesting the

possibility of a weak electrostatic interaction between the two naphthalene chromophores. We believe that such a small electrostatic interaction is important to realize the unique time-evolving supramolecular polymerization.

Specific Comments (2):

In a manuscript dealing with co-assembly, it is very important to derive the thermodynamic parameters associated to the supramolecular polymerization of both the pristine components and also to the investigated mixture of components. The authors have made such calculations and provide the corresponding parameters. However, there are two circumstances that make these values not very accurate. The first one is the fact the authors have utilized the model described by Meijer et al. in 2006 (reference S9 in the Supporting Information). At the same time, and more importantly, the authors have utilized only three curves to derive these parameters. Of course, if you have three point the R2 value for the linear fitting is 1. The authors should: a) make the calculations with, at least, four cooling curves or b) utilize the most recent model published by ten Eikelder and coworkers (J. Phys. Chem. B, 2012, 116, 5291) that allows a global fitting of the data and, consequently, a more accurate determination of the parameters.

Author's Response to the Specific Comments (2): Thank you very much for this very important suggestion. According to the suggestion, we have revised the van't Hoff analysis by further adding the data recorded at $c_t = 100 \mu\text{M}$ for **1** and **2** (Supplementary Figure 7) and $c_t = 74$

μM for the 1:1 mixture (Supplementary Figure 9). As shown below, the new data are also in line with the data previously measured for other concentrations. Accordingly we provided these new results with the revised Figure 6f–h, Supplementary Figure 7 and 9, and revised the corresponding parameters in Supplementary Table 1. The main text (pages 5, 6, 12, and 13) has been also revised accordingly.

Original Figure 6f–h:

Revised Figure 6f–h:

Original Supplementary Figure 7:

Supplementary Figure 7.

Plots of the degree of aggregation of **1** (a) and **2** (b) in MCH at different concentrations ($c_i = 30 \mu\text{M}$, $40 \mu\text{M}$, and $50 \mu\text{M}$) calculated from the absorption change at $\lambda = 470 \text{ nm}$ as a function of the temperature during the cooling process (cooling rate: 1 K min^{-1}).

Revised Supplementary Figure 7:

Supplementary Figure 7.

Plots of the degree of aggregation of **1** (a) and **2** (b) in MCH at different concentrations ($c_i = 30 \mu\text{M}$, $40 \mu\text{M}$, $50 \mu\text{M}$, and $100 \mu\text{M}$) calculated from the absorption change at $\lambda = 470 \text{ nm}$ as a function of the temperature during the cooling process (cooling rate: 1 K min^{-1}).

Original Supplementary Figure 9:

Supplementary Figure 9.

Plots of the degree of aggregation of $\mathbf{1/2}$ (1:1) in MCH at different concentrations ($c_t = 60 \mu\text{M}$, $80 \mu\text{M}$, and $100 \mu\text{M}$) as a function of temperature during the cooling process (cooling rate = 1 K min^{-1}).

Revised Supplementary Figure 9:

Supplementary Figure 9.

Plots of the degree of aggregation of $\mathbf{1/2}$ (1:1) in MCH at different concentrations ($c_t = 60 \mu\text{M}$, $74 \mu\text{M}$, $80 \mu\text{M}$, and $100 \mu\text{M}$) as a function of temperature during the cooling process (cooling rate = 1 K min^{-1}).

Original Supplementary Table 1:

Supplementary Table 1.

The standard enthalpy (ΔH°), entropy (ΔS°), and Gibbs free energy (ΔG°) of **1**, **2** and **1/2** obtained by a van't Hoff analysis in Supplementary Figs 7, 9, 28.

	ΔH° (kJ mol ⁻¹)	ΔS° (J mol ⁻¹ K ⁻¹)	ΔG° (kJ mol ⁻¹) at 298 K
1	-63.0	-110.2	-30.2
2	-53.6	-78.7	-30.2
1/2 (amorphous coaggregate)	-56.9	-93.0	-29.7
1/2 (helicoid)	-95.9	-217.8	-31.9

Revised Supplementary Table 1:

Supplementary Table 1.

The standard enthalpy (ΔH°), entropy (ΔS°), and Gibbs free energy (ΔG°) of **1**, **2** and **1/2** obtained by a van't Hoff analysis in Supplementary Figs 7, 9, 28.

	ΔH° (kJ mol ⁻¹)	ΔS° (J mol ⁻¹ K ⁻¹)	ΔG° (kJ mol ⁻¹) at 298 K
1	-72	-139	-31
2	-58	-92	-31
1/2 (amorphous coaggregate)	-55	-88	-29
1/2 (helicoid)	-96	-218	-32

Page 14

Original:

From these ΔH° and ΔS° values, the T_c at which $\Delta G^\circ_{\text{amo}} = \Delta G^\circ_{\text{hel}}$ was calculated to be 39 °C (Fig. 6f).

Revised:

From these ΔH° and ΔS° values, the T_c at which $\Delta G^\circ_{\text{amo}} = \Delta G^\circ_{\text{hel}}$ was calculated to be 42 °C (Fig. 6f).

Specific Comments (3):

The authors report on a very complex system composed by three components, in certain cases one of them is chiral. It is absolutely necessary to perform the corresponding studies with binary systems. For instance, how are the binary 1/2 or 2/3 systems behaving? More interesting is the ternary systems incorporating the chiral congener. In this case, the mixtures 1+1S or 2+2S would be examples of sergeant-and-soldiers experiments that provide very useful information about the required ratio of the chiral sergeant, and also the energetics derived of this chiral amplification phenomenon, to further optimize the investigation of the ternary systems.

Author's Response to the Specific Comments (3): We partially showed the sergeant-and-soldiers experiment of the binary and ternary system because of the out of the scope of the main topic. We are currently on-going comprehensive study with also use of (*R*)-derivatives, are happy to update our reports in the near future.

REVIEWERS' COMMENTS:

Reviewer #1 (Remarks to the Author):

The manuscript has been revised very carefully according to the reviewer's comments. All the questions of the reviewers have been well addressed. I think the current version of manuscript meets the criteria for Nature Comm. and I recommend the publication.

Reviewer #2 (Remarks to the Author):

I have carefully read the responses to the referees made by the authors. I believe that the authors have addressed the concerns raised by the reviewers. I, therefore, recommend the acceptance of the manuscript.

Reviewer #3 (Remarks to the Author):

The authors have addressed appropriately all the concerns raised by the three referees and, especially, to the comments made by me. I consider that the revised version of the manuscript, together with the explanations provided by the authors in the rebuttal letter, confirm that the manuscript meets all the quality and novelty criteria on Nature Communications to be published in this journal.

Therefore, I consider that the manuscript is suitable for publication in Nature Communications in the present form